# Comparing Bayesian-Based Reconstruction Strategies in Topology-Based Pathway Enrichment Analysis

**DOI:** 10.3390/biom12070906

**Published:** 2022-06-28

**Authors:** Yajunzi Wang, Jing Li, Daiyun Huang, Yang Hao, Bo Li, Kai Wang, Boya Chen, Ting Li, Xin Liu

**Affiliations:** 1Department of Biological Sciences, School of Science, Xi’an Jiaotong-Liverpool University, Suzhou 215123, China; yajunzi.wang17@student.xjtlu.edu.cn (Y.W.); daiyun.huang18@student.xjtlu.edu.cn (D.H.); ting.li01@xjtlu.edu.cn (T.L.); 2Wisdom Lake Academy of Pharmacy, Xi’an Jiaotong-Liverpool University, Suzhou 215123, China; jing.li1902@student.xjtlu.edu.cn (J.L.); yang.hao20@student.xjtlu.edu.cn (Y.H.); bo.li21@student.xjtlu.edu.cn (B.L.); kai.wang17@student.xjtlu.edu.cn (K.W.); boya.chen16@student.xjtlu.edu.cn (B.C.)

**Keywords:** topology-based pathway analysis, Bayesian network, network reconstruction, gene expression

## Abstract

The development of high-throughput omics technologies has enabled the quantification of vast amounts of genes and gene products in the whole genome. Pathway enrichment analysis (PEA) provides an intuitive solution for extracting biological insights from massive amounts of data. Topology-based pathway analysis (TPA) represents the latest generation of PEA methods, which exploit pathway topology in addition to lists of differentially expressed genes and their expression profiles. A subset of these TPA methods, such as BPA, BNrich, and PROPS, reconstruct pathway structures by training Bayesian networks (BNs) from canonical biological pathways, providing superior representations that explain causal relationships between genes. However, these methods have never been compared for their differences in the PEA and their different topology reconstruction strategies. In this study, we aim to compare the BN reconstruction strategies of the BPA, BNrich, PROPS, Clipper, and Ensemble methods and their PEA and performance on tumor and non-tumor classification based on gene expression data. Our results indicate that they performed equally well in distinguishing tumor and non-tumor samples (AUC > 0.95) yet with a varying ranking of pathways, which can be attributed to the different BN structures resulting from the different cyclic structure removal strategies. This can be clearly seen from the reconstructed JAK-STAT networks by different strategies. In a nutshell, BNrich, which relies on expert intervention to remove loops and cyclic structures, produces BNs that best fit the biological facts. The plausibility of the Clipper strategy can also be partially explained by intuitive biological rules and theorems. Our results may offer an informed reference for the proper method for a given data analysis task.

## 1. Introduction

The development of high-throughput omics technologies has revolutionized the study of life. Often the quantification of the whole genome is just a first step in generating hypotheses based on differentially expressed genes (DEGs) and pathways. Pathway enrichment analysis (PEA) offers a set of convenient solutions for the dimensionality reduction of massive data and the systematic interpretation of the functional relations of the predefined subsets of biomolecules [1]. They are often applied to gain a definite and limited set of DEGs and associated intergenic relations through a pool of omics data [1].

Chronologically, the past two decades have seen three generations of PEA approaches (Table 1): over-representation analysis (ORA), functional class scoring (FCS), and topology-based pathway analysis (TPA) [1]. The first generation, ORA, measures the fraction of a subset of DEGs enriched in specific pathways [2]. Despite being straightforward and efficient, methods such as GoMiner [3] and WebGestalt [4] are somewhat arbitrary in DEG selection, which may yield different results. In addition, continuous expression changes of genes are ignored in these methods, i.e., each gene was seen as an independent entity in the system, making these methods unable to assign different weights to input genes, or to reflect the regulatory changes between genes [2]. The second generation, FCS, with the gene set enrichment analysis (GSEA; [5]) as the most popular representative, starts to combine coordinated changes in the expression of genes to evaluate their enrichment in specific pathways [2]. FCS methods address the limitations of ORA. However, the pathway topology and biological causal relations between genes are still omitted in this generation of methods [5], compromising their insights into the systems-level understanding of biological networks and interpretability [6], arousing the third generation of PEA methods. The third generation, TPA methods, cope with the limitations of ORA and FCS by the integration of pathway topology, enabling a more precise assessment of the statistical relevance of both genes and pathways, as well as the deciphering of biological causal relations between pathway components [7]. Compared to ORA and FCS, TPA generally provides improved sensitivity and specificity and better pathway ranking in most cases [7].

Since the first statistical approach to scoring changes in the activity of metabolic pathways from gene expression data [16], a number of TPA methods have been introduced. A thorough review and comparison of these methods can be found elsewhere [17,18]. Some of these methods apply the analysis to binary interaction networks built from the input pathway, while some others transform the pathways into graphs using different strategies [17]. An example of the latter is TopologyGSA, which transforms the directed input pathways into undirected graphs by connecting the parents of each child and removing the edge direction [12].

Another set of TPA methods, such as BPA [13], BAPA-IGGFD [14], and BNrich [9], reconstruct the preprocessed pathways by training Bayesian networks (BNs). Being a probabilistic graphical model, a BN provides arguments in an uncertainty framework by a directed acyclic graph (DAG) to indicate the conditional dependencies (edges) between variables (nodes) [19]. BNs, as powerful networks for integrating expression data with comprehensive causal relationships, have been shown to be ideal topology structures to explain potential connections between genes [19].

The structures of real biological pathways are complicated, often consisting of loops or cyclic structures. The existence of cycles in the pathway networks conflicts with the requirement of DAGs in BNs [13]. Therefore, in most cases, reconstructing pathway networks to eliminate cyclic structures is an indispensable step when constructing BNs [13]. In fact, the BN structure reconstruction strategies employed by distinct BN-based TPA methods are significantly different. Specifically, BPA models pathways as BNs by merging repeating components and uses Spirtes’ method [20] to eliminate cyclic structures [13]. BAPA-IGGFD rebuilds the pathway graphs by retaining edges representing activation and inhibition while removing those representing interactions [14]. BNrich uses a set of biological intuitive rules to eliminate cyclic structure and employs LASSO to simplify the BN structure [9]. PROPS [11] is the most recent generation Gaussian BN-based TPA software package for disease classification. It adds edges in a random order to reconstruct BNs and uses gene expression data to perform probabilistic scoring to input KEGG pathways [11].

In a recent study by Ihnatova et al., seven non-BN-based TPA methods were critically assessed with respect to their performance in identifying differentially expressed pathways between two contrary groups from gene expression data [17]. However, little effort has been devoted to comparing the difference in the PEA performance of BN-based TPA methods due to the use of different topology reconstruction strategies. Therefore, a comprehensive comparison of these methods is urgently needed. In addition to BPA, BNrich and PRPOS, a non-BN-based TPA method, Clipper [10], was also included in the comparison as it converts pathways into DAGs before creating junction trees. In particular, Clipper implements a linear regression for each parent–child node pair and uses the significance of each edge as the strength of evidence to rank edges. The weakest one among the edges constructing the cyclic structure is removed. Therefore, by extracting its reconstruction method and following the same procedure in parameter learning and downstream analysis, we can treat it as a BN-based method that can be compared with the other strategies. KEGG2Net is a newly developed web portal that offers multiple strategies to convert biological pathways into DAGs [21]. Among the four alternative converting ways, the Ensemble method [15], which uses a Bayesian skill rating system and a social agony metric to infer graph hierarchy, finds the best DAG that explains the causal relations between genes [21]. Therefore, the Ensemble method was also incorporated in this comparative study. BAPA-IGGFD was omitted in this comparison due to its inaccessibility.

Existing BN-based TPA approaches evaluate their PEA results and their performance in a disease state classification based on disparate gene expression data, making fair comparisons between the various approaches difficult. Moreover, the biological relevance of the reconstructed BNs is insufficiently or barely disclosed. In light of these perceived needs, we collected five sets of gene expression data from Gene Expression Omnibus (GEO) and The Cancer Genome Atlas (TCGA) containing at least 30 paired tumor and adjacent non-tumor samples, HCC, HNSC, THCA, KIRC, and LUAD, and performed PEA and disease state classification. The decision tree-based Random Forest was employed to calculate pathway rankings in the classification contribution. The collection of biomolecules and their causal relations of JAK-STAT signaling, a canonical pathway with complex negative feedback loop structures, was used to illustrate the differences of these approaches in the reconstructed BN structures and their relevance to the biological facts.

As a result, BPA, BNrich, PROPS, Clipper, and Ensemble performed equally well in distinguishing tumor and non-tumor samples (AUC > 0.95) across all five datasets. However, their ranking of pathways by feature importance varies widely, which can be attributed to the different BN structures resulting from the different cyclic structure removal strategies. By involving intuitive biological rules, BNrich produces the best BNs that preserve causal relations between genes that best fit the biological facts, and the most reasonable ranking of enriched pathways. Clipper measures edge importance and preserves edges on the assumptions of linearity and conditional independence that are often not met in biology. Nonetheless, the plausibility of Clipper can be partially explained by the dominant signaling and transition state theorem in biology. Despite the excellent classification performance, the randomness of PROPS and BPA hamper their biological interpretability, as does the EN method due to its graph hierarchy-based strategy, which ignores gene expression quantification and biological knowledge.

To summarize, this is the first comprehensive study that assesses the most recent and widely used BN-based strategies for biological pathway reconstruction. Their PEA and disease state classification performance were evaluated and compared on five benchmark gene expression datasets. A canonical pathway, JAK-STAT signaling, was used to illustrate the differences of the reconstructed BNs and their biological relevance. The advantages and limitations of each strategy were also thoroughly discussed, prompting possible future research directions for an optimized network reconstruction strategy.

## 2. Materials and Methods

### 2.1. Dataset Pre-Processing

Five datasets quantifying the gene expressions of paired tumor and adjacent non-tumor samples from multiple cancer types—hepatocellular carcinoma (HCC, GSE144269, *n* = 140), head and neck squamous cell carcinoma (HNSC, *n* = 86), kidney renal clear cell carcinoma (KIRC, *n* = 142), lung adenocarcinoma (LUAD, *n* = 112), and thyroid carcinoma (THCA, *n* = 112)—were downloaded from GEO and TCGA to construct sets of benchmark data for the fair evaluation and comparison of reconstructed BNs using different strategies. RNA sequencing expression read counts downloaded from TCGA were normalized in FPKM (fragments per kilobase of transcript per million mapped reads). The processed expression levels were subject to log2 (*n* + 1) transformation. The paired samples in the dataset were randomly split into training (75%) and test (25%) sets ten times. Only data from the training set were utilized for BN reconstruction, BN parameter learning, and classification model training.

Kyoto Encyclopedia of Genes and Genomes (KEGG), among the most authoritative pathway databases, adds value to a systematic understanding of complex biological systems by providing a shared resource of known pathways. In total, 187 and 270 KEGG pathways are implemented in BNrich and PROPS, respectively. Overlapped pathways with more than five edges after reconstruction were preserved, resulting in 157 pathways that contribute to the downstream analysis.

### 2.2. Reconstruction of BN Structures

The first step in building a BN is to reconstruct the pathways using directed acyclic graphs (DAG). In our comparison, we used the preprocessed datasets to reconstruct the BN structure, depending on the underlying structure of the selected KEGG pathway. Five different reconstruction methods were incorporated in this study, including BPA, PROPS, Clipper, BNrich, and Ensemble method. The detailed strategies of these approaches are illustrated in the following subsections.

#### 2.2.1. Reconstruction of BN Structures by BPA

BPA is one of the earliest efforts to model pathways as BNs. In BPA, cycles are not handled individually but are considered in groups. Specifically, sets of cycles sharing at least one node are first found using Tarjan’s algorithm and termed cyclegroups. Then, for each cyclegroup, all edges between nodes are removed. The nodes within a specific cyclegroup are randomly numbered without duplication. Directed edges pointing from the smaller numbered nodes to the larger numbered nodes are then added. In addition, for parent node(s) that are outside the cyclegroup, directed edges pointing to child node(s) are also added (see also Appendix A). Such remedies can reduce the loss of information caused by cyclic edge removal as much as possible, but it will inevitably add a large number of redundant edges.

#### 2.2.2. Reconstruction of BN Structures by PROPS

In PROPS, KEGG human pathways are extracted as reference pathways using the *KEGGgraph* R package. For each selected path, nodes (genes) and associated edges not included in the preprocessed dataset are removed. To convert the original pathway into a DAG for BN modeling, an empty graph is created with all genes in the pathway as nodes, and an edge is randomly selected from the pathway and added to the graph each time unless a cycle is triggered (see also Appendix A). The edge that can lead to cycles was detected by the built-in function check.cycles in the *bnlearn* R package (version 4.7.1). Finally, pathway networks with more than five edges remain in the analysis.

#### 2.2.3. Reconstruction of BN Structures by Clipper

Like PROPS, in Clipper, KEGG human pathways are obtained from the *KEGGgraph* R package, and genes not included in the preprocessed datasets are discarded. Instead of randomly removing edges that can form a cycle, Clipper employs the strength of linear relationships between nodes inferred from gene expression data to guide edge elimination. To derive a measure of the correlation between nodes connected by each edge, linear regression models are constructed using gene expression data with parent nodes as explanatory variables and child nodes as response variables. Specifically, the *p*-value of a standard significance test with zero coefficients is used as the strength of evidence. The list of edges is then sorted by their corresponding *p*-values from small (significant) to large (not significant). Finally, edges that do not generate cycles (checked by the built-in function in the *bnlearn* package) are sequentially added to the graph so that the removed edges represent minimal expression profile correlations (see also Appendix A). Likewise, we keep pathway networks with more than five edges.

#### 2.2.4. Reconstruction of BN Structures by BNrich

In BNrich, 187 human non-metabolic pathways are selected from the KEGG pathway. Unlike other methods, BNrich eliminates cyclic structure based on biological principles rather than computational methods. Elimination rules focus on edges connecting gene products, feedbacks in cytoplasmic processes originating from the nucleus, and the main direction of signaling transmission from the cell membrane to the nucleus (see stage 1 in Appendix A).

With the help of this biological knowledge, all pathways are transformed into a directed acyclic graph. In addition to removing edges in the network that cause cycle structures, BNrich also simplifies the structure by removing edges that cannot be supported by expression profile correlations. Specifically, LASSO regression [22] is used as a variable selection method, where the parameters are estimated according to the following formulas
(1)$$\hat{\beta }\left(\lambda \right)=argmin_{\beta }\left(\frac{{\left\|Y-X\beta \right\|}_{2}^{2}}{n}\right)+\lambda {\left\|\beta \right\|}_{1},$$
(2)$${\left\|Y-X\beta \right\|}_{2}^{2}=\sum_{i=0}^{n}{\left(Y_{i}-{\left(X\beta \right)}_{i}\right)}^{2}\;\mathrm{and}\;{\left\|\beta \right\|}_{1}=\sum_{j=1}^{k}\beta_{j}$$
where λ is the penalty level, a larger λ means a greater amount of shrinkage.

For each parent–child node pair in the acyclic pathway graph, BNrich fits two LASSO regression models using tumor and non-tumor samples, respectively. Parent nodes are treated as explanatory variables, and child nodes are response variables. Relying on the L1 penalty, LASSO shrinks the coefficient estimates for weak edges to zeros. Therefore, the edge with the minimum expression profile correlation should be assigned zero coefficients on both samples and thus be removed from the pathway graph to simplify the structure. In practice, the minimum mean error of the cross-validation is used to help obtain the best λ value. Finally, reconstructed pathway networks with more than five edges are retained in the analysis.

#### 2.2.5. Reconstruction of BN Structures by Ensemble Method

Ensemble method attempts to break cycles from a directed graph while preserving its graph hierarchy (logical structure) as much as possible [15]. First, the graph hierarchy is inferred through a Bayesian skill rating system (*TrueSkill*) and a social agony metric, generating two hierarchical ranking scores for each node. TrueSkill treats every node as a player and every edge as a competition. The skill level of each node is formulated as normal distribution and updated according to the edge direction. If there is an edge from $v_{1}$ to $v_{2}$, then player $v_{2}$ is considered to win against $v_{1}$, and this outcome is compared with the expected outcome to update skill levels of $v_{1}$ and $v_{2}$. In social agony metric, an edge from $v_{1}$ to $v_{2}$ means a recommendation from $v_{1}$ to $v_{2}$. If there exists no reverse edge, $v_{2}$ is considered to be higher ranked than $v_{1}$ because of the assumption that higher-ranked nodes are less likely to connect to lower-ranked nodes. When an assumption is violated (an edge from a higher-ranked node to a lower-ranked node), it causes social agony, which is defined by the difference between the ranks of two nodes. The final ranking of each node is inferred by minimizing the total agony of the graph.

Then, an iterative process of simplification and edge removal with three heuristic strategies (forward, backward, and greedy) is applied to the directed graph until the graph becomes acyclic. Since there are two hierarchical ranking metrics and three edge removal strategies, combining them in pairs yields six decycling methods. In practice, the process first divides a pathway graph into strongly connected components (SCCs) using Tarjan’s algorithm (implemented in the Python package *networkx*) and then decycles the SCCs using all six methods independently. The process is iteratively applied to resulting SCCs until no more SCCs can be generated (see also Appendix A). The final score for an edge is produced from an ensemble of all six methods
$$\underset{m}{\sum }I_{m}\left(e\right)$$
where m is the method and $I_{m}\left(e\right)$ indicates whether to remove (=1) or keep (=0) edge e by method m.

### 2.3. BN Parameter Learning by Non-Tumor Data

With the refined path DAG structure, the next step is to use the expression data for parameter learning of the Bayesian network. Considering the tumor vs. non-tumor classification in downstream analysis, only non-tumor gene expression data is used to learn the parameters. Each KEGG pathway is modeled by a Gaussian Bayesian network, where a gene node represents gene expression, and each node is fitted using a linear combination of its parent nodes. For example, if a children node Y has n parent nodes $X=\left\{X_{1},X_{2},\cdots ,X_{n}\right\}$, their linear relationship is formulated as
(3)$$Y=\beta_{0}+\sum_{i=1}^{n}\beta_{i}X_{i}+\varepsilon ,$$
where the coefficients $\beta_{i},\;i=1,2,\cdots n$ are optimized by Maximum Likelihood Estimation (MLE), a common parameter learning method that estimates parameters by maximizing a likelihood function based on known formulas and given data. In practice, parameter learning is implemented using the function bn.fit from the R package *bnlearn*.

### 2.4. Pathway-Based Feature Engineering

Ideally, pathway networks that learn parameters from non-tumor data should yield distinguishable likelihoods when inferring non-tumor and tumor samples, reflecting the corresponding pathway activity. Therefore, the log-likelihood value of each pathway BN for each patient should serve as a robust feature for building classification models for tumors and non-tumors. The log-likelihood is calculated through the following formula
(4)$$\log P(X_{1}=x_{1},\;\ldots ,X_{n}=x_{n}\;|\;\theta )=\sum_{i-1}^{n}\log P(X_{i}=x_{i}\;|\;\theta ,\;X_{pa}=x_{pa}),$$
where $X=\left\{X_{1},X_{2},\cdots ,X_{n}\right\}$ are nodes in a specific pathway; $x=\left\{x_{1},x_{2},\cdots ,x_{n}\right\}$ are observed RNA-seq data; θ is the learned parameters from non-tumor data; and $X_{pa}$ refers to the parents of node $X_{i}$. The R function *logLik* is used to implement log-likelihood calculation. Together, log-likelihood values of all used pathways for each sample are sent to Random Forest model to train a binary classifier. To avoid data leakage, parameter learning for each pathway network is performed using only non-tumor data in the training dataset.

### 2.5. Pathway Enrichment through Random Forest Classification

Following PROPS, we use the log-likelihood values from the reconstructed pathway BNs for tumor and non-tumor classification and exploit feature importance for pathway enrichment analysis. The hyperparameters ntree (number of trees used in aggregation) and mtry (number of variables available for splitting at each tree node) are optimized by grid search. The model equipped with optimized hyperparameters is used to perform training and prediction. The performance on the test set is reported in terms of accuracy, precision, recall, F1-score, and the area under the receiver operating characteristic curve (AUC). Except for the AUC, the threshold is determined by the best Matthews correlation coefficient. Feature importance is assessed by mean drop Gini (MDG), which should be a useful value to quantify and rank the enrichment scores for each pathway.

## 3. Results

### 3.1. Different BN Reconstruction Strategies Lead to Different PEA Results

The random divisions of paired samples into training and test sets were repeated ten times, and used for downstream PEA using five methods, BNrich, PROPS, Clipper, BPA, and Ensemble method, respectively. All five methods were generally equally good at distinguishing between tumor and non-tumor samples using log-likelihood values of their respective reconstructed BNs. The classification performance on each dataset reveals slight differences with AUC 0.95–0.97 on HCC and HNSC, 0.97–0.98 on THCA, and 0.99–1.0 on KIRC and LUAD (Table 2).

Although all five approaches produced consistently high accuracy in the PEA-based classification of tumor and non-tumor samples, there were significant differences in the rankings of pathways enriched by the five methods according to the feature importance based on MDG scores (Appendix A). Figure 1 summarizes the top ten pathways enriched by any of the five methods in the HCC dataset. Some pathways, including focal adhesions, PPAR signaling, ECM-receptor interactions, and chemical carcinogenesis, consistently ranked high in the PEA results across the five methods. However, we also observed some pathways, such as basal cell carcinoma, the JAK-STAT signaling pathway, Rap1 signaling pathway, and regulation of actin cytoskeleton, were assigned extremely low feature importance by MDG scores by at least one method. Since key factors such as gene expression data, the collection of pathway components and causal relationships, feature engineering, and feature importance scoring methods involved in this comparison remain unchanged, it is reasonable to speculate that the differences in the rankings lies in the different BN reconstruction strategies used in the five methods. Inspired by these observations, we intend to explore the topological structures of the BNs reconstructed by the five methods and their agreement with the molecular biological observations. Pathways with varying rankings among the five methods, especially those with cyclic structures such as the JAK-STAT signaling pathway, are ideal candidates for such a comparison. For PEA results on the other tumor datasets, please refer to the Appendix A.

### 3.2. Different Strategies Generate Different BN Structures

The main challenge in modeling pathways using BNs is to remove cyclic structures while preserving key biological context. In this study, we investigate five methods for determining the edges to be eliminated, BNrich, PROPS, Clipper, BPA, and Ensemble method. These five approaches represent four main directions, based on randomness, gene correlations inferred from expression data, graph hierarchies, and biological intuitive rules (Figure 2).

PROPS provides the most straightforward method. For those edges that form a cyclic structure, PROPS assumes to be based on no prior knowledge about the edges, so every edge has an equal probability of being dropped (Figure 2a). This approach is simple and effective, but the resulting BNs may deviate from biological facts or principles. This is especially true when looking at the averaged ranks of Bayesian information criterion (BIC) scores of all reconstructed pathway networks on non-tumor samples (Table 3). PROPS consistently ranks in the bottom two positions in all datasets, indicating that randomness can indeed disrupt biological network hierarchies or lose important causal relationships between genes, which lead to the hindered interpretability of this model structure. In summary, in an implementation with randomness on HCC, PROPS eliminated 601 edges from 11,301 edges (43 pathways), of which 115 edges cause self-loops and 486 edges come from cyclic structures, resulting in 5.32% edges loss.

It is worth noting that BPA (Figure 2e), one of the first approaches to use BNs for pathway analysis, can also fall in this direction and randomly drops edges in cycles. In addition to eliminating edges, one unique characteristic of BPA lies in that it also adds new edges to the graph in order to preserve, as much as possible, the relationships broken by the removed edges while maintaining conditional independence of the graph. BPA groups cycles that share at least one common node and follows two principles for adding edges: nodes from different cycles in one group can be connected in a random direction; for each parent node outside a specific cyclegroup, a directed edge from that parent node to each node within the cyclegroup is added to the graph. In an implementation with randomness on HCC (11,301 edges/43 pathways), BPA removed 862 edges in addition to those that cause self-loops, but on the other hand, it added a total of 8281 edges that were not originally included in KEGG pathways. The edge compensation strategy applied in the BPA pathway may serve as a guide for subsequent method development to reduce the loss of biological causality upon reconstruction and preserve intact upstream and downstream relationships as much as possible.

On the other hand, Clipper (Figure 2b) computationally learns expression profile correlations to measure the strength of evidence for each edge when sufficient RNA-seq samples are available. Linear regression was performed between each node and its parent node using normalized expression data from non-HCC samples. Therefore, a significance test using the zero-parameter hypothesis provides evidence supporting the corresponding edge. While the weakest edges can be removed based on parameter learning, the measure of strength depends heavily on the data samples used and the assumptions of linearity and conditional independence in the modeling. Starting from the same source annotation, Clipper dropped 544 edges in order to remove the cyclic structure. Counting 128 more self-loop edges, a total of 5.83% of the edges were removed from the pathway. Although PROPS and Clipper removed a close number of edges, only about half of the eliminated edges were removed by both methods.

The Ensemble method (Figure 2d) also computationally seeks suitable edges that break cycles, but instead of learning evidence from gene expression data, it makes decisions based on graph hierarchies of input pathways. The main mechanism behind this approach is the assumption that the node receiving an edge should be ranked higher than the node giving it. Under such rules, genes with more upstream genes and fewer downstream genes in the pathway are less likely to be linked with genes with more downstream genes. After iteratively updating the rank of genes, the edge between the genes with the largest rank gap and causing the cycle will be voted to be removed. On the HCC dataset, the Ensemble method removed 456 edges, which is less than PROPS (random) and Clipper (learn from data).

Unlike Clipper and Ensemble, BNrich takes a biological approach rather than relying on computation to remove cyclic structures (Figure 2c). With expert knowledge of cytoplasmic processes and the direction of signaling flux, BNrich manually refines the KEGG signaling pathway to create DAGs for Bayesian modeling. Although it is reasonable and reliable, it requires extra careful work in the study of new pathways. After the first stage of reconstruction, for the 43 cyclic pathways we studied, 850 edges were dropped. In addition to reconstructing the structure of BN, BNrich also considers simplifying the network by removing weak edges using a method similar to that used in Clipper (regression between pairs of parent–child nodes), even though these edges do not participate in forming the cyclic structure. Specifically, LASSO regression is applied, which shrinks the parameter of insignificant edge to zero. This simplification could, in theory, reduce analysis complexity without losing vital information when assumptions are fulfilled. However, whether such edge pruning would impair important biological causal relationships should be critically assessed in downstream pathway analysis. Compared to stage 1, stage 2 simplification eliminated 1380 additional weak edges. It is worth noting that this stage 2 simplification can be applied to DAGs reconstructed by any method, including the other four compared approaches. Additionally, a similar effect can be achieved by eliminating insignificant edges defined using a preset *p*-value threshold (e.g., *p*-value ≥ 0.05) in Clipper.

### 3.3. Reconstructed JAK-STAT Networks and Biological Relevance

The reconstructed BN structures of the JAK-STAT signaling pathway (Figure 3) clearly illustrate the differences in the BN reconstruction strategies between methods incorporated in this study. Janus tyrosine kinase (JAK) is a family of proteins that function as intracellular non-receptor tyrosine kinases containing four members JAK1, JAK2, JAK3, and TYK2. Signal transducers and activators of transcription (STAT), consisting of seven subtypes STAT1, STAT2, STAT3, STAT4, STAT5a, STAT5b, and STAT6, is a group of transcription factors activated by JAK [23]. The JAK-STAT signaling is an evolutionarily conserved pathway that plays critical roles in signal transduction cascades [24]. The dysregulation of JAK-STAT signaling was found to be associated with various infectious diseases [25], immune system disorders [26], and oncogenesis and proliferation [27]. The activation of JAK-STAT signaling starts with the association of cytokines or growth factors and transmembrane receptors. Noncovalent bound to cytokine receptors, JAKs mediate tyrosine phosphorylation of the receptors and recruit STATs. Phosphorylated STATs dimerize, ultimately leading to their nuclear translocation and DNA binding to regulate gene expression in the cell nucleus [28], including suppressors of the cytokine signaling (SOCS) family of proteins, which serve as negative feedback inhibitors of JAK-STAT [29].

In the case of BNrich, it was not surprising to see that all the negative feedback edges representing SOCS inhibition of JAK tyrosine activity and JAK-STAT signal transduction were removed (Figure 3). Such a strategy is in line with the biological principle that, in most cases, signal transduction is the transmission of molecular signals from the outside of the cell to the inside. Negative feedback occurs when the original effect of the stimulus is reduced by the output, restoring biological systems back to homeostasis, which is often disrupted in tumor cells. From this perspective, preserving as many activation edges as possible may improve the interpretability of enriched pathways containing negative feedback structures and the accuracy of using them for sample classification.

The same removal of SOCS negative feedback edges was observed in the case of the Ensemble method, which can be explained by the nature of the Ensemble strategy. A signaling pathway is an artificially defined subset of chemical reactions in a cell that lead to a certain product or a change. In the spectrum of KEGG-defined JAK-STAT signaling, SOCS expression is regulated by the DNA binding of STAT proteins [30]. JAK-STATs also act as key regulators of many other genes’ expression and downstream Ras-Raf and PI3K-AKT signalings, making them relatively low-ranked among JAK-STAT signaling genes according to *TrueSkill* hierarchical ranking scores. Given the assumption of the Ensemble method, which prioritizes maintaining a network hierarchy that minimizes social agony, it is plausible that the negative feedback edges from SOCS to JAK-STATs were removed.

Compared to BNrich and Ensemble, Clipper removed more edges representing activated JAKs and/or the cytokine receptor phosphorylation of STATs (Appendix A). While this seems to go against the well-known static JAK-STAT network topology, signal transduction in cells is, in fact, an adaptive process, i.e., in the case of stimuli, interconnected genes will eventually develop a most effective path, largely via which the extracellular signals will be transferred into the nucleus [31,32,33]. Clipper measures edge importance in terms of linear correlations between adjacent nodes. It is probable such measurements aided in the location of so-called dominant signalings [34,35]. However, it is worth noting that this linearity is often not met due to sample heterogeneity and the presence of bypath and loop structures in pathways. To test our hypothesis, we randomly shuffled gene expression values across multiple samples. As a result, the BN reconstructed from the shuffled dataset has a topological structure that is significantly different from the one from the real expression data, which is more similar to a random reorganization of the gene relationships from the input pathway (Figure 4).

Another difference in the Clipper- and Ensemble-reconstructed BNs is that they retained a number of edges representing state transitions (Appendix A). Oncogenesis can be seen as a transition from the healthy state to the disease state [36]. Cells function relying on the time-sequential cascade reactions among massive biomolecules in them. Transition state theory, which assumes a quasi-equilibrium between the reactants and the products and explains the reaction rates of chemical reactions [37], has been applied to model the disease state by the observations of composition changes based on gene expression and the generated hypothesis about the critical points and key features of disease development [36,38]. Consistent with this theory, preserving these state transition edges may constitute the topological basis for disease state transition modeling, providing additional insights into the discovery of key genes or pathways associated with disease development or sample classification.

PROPS and BPA reconstruct BNs by adopting a randomness strategy. We were curious to what extent this randomness would affect the reconstructed network structure and whether it would impair downstream pathway analysis. To this end, we randomly generated ten reconstructed BNs and compared them in the case of PROPS and BPA, respectively (Appendix A). Figure 5 shows the superimposition of the two randomly selected reconstructed networks and highlights their differences. It can be seen that the coincidences in the superimposed networks are only less than 40% in the case of PROPS and even worse in the case of BPA. More importantly, some important biologically-confirmed edges were arbitrarily omitted. Despite their good classification performance, the randomness of PROPS- and BPA-generated BNs may hamper their biological interpretability. It is also worth noting that, unlike PROPS, in addition to removing cycle-forming edges, BPA adds almost the same number or even more edges as a remedy. While allowing genes to be linked in another way as compensation, it is obvious that some important direct causal relationships between genes may be dropped, while the discovery of some true biological relevance may be hindered by redundant edges; spurious indirect relationships may also proliferate.

## 4. Discussion

In this section, we demonstrate the pros and cons of each approach based on a discussion of algorithmic differences, the experimental results on five gene expression datasets, and the reconstructed structures of the JAK-STAT signaling pathway. Given the apparent complementarity, we present a potential method combining existing methods as a suggestion for future research. In addition, potential links between the BN reconstruction of pathways and learning of the BN structure from gene expression data are discussed.

### 4.1. Concluding Remarks

Bayesian networks represent a major class of methods for performing topology-based pathway analysis, the latest generation of pathway enrichment analysis methods. While BN allows incorporating pathway topology for global pathway analysis, it requires the input network to be directed and acyclic. Therefore, eliminating cyclic structures from the experimentally identified pathways becomes a first and crucial step. In this study, we rigorously compared five different strategies for reconstructing pathways into BNs proposed by the BPA, PROPS, Clipper, BNrich, and Ensemble method, respectively. These strategies fall into four main directions to remove edges: random selection, linear relationship strength learned from expression data, biological intuitive rules, and graph hierarchies. In particular, in addition to BN reconstruction, BNrich also applies LASSO regression to discard edges whose linear correlations cannot be supported by the expression data, to remove redundant edges and simplify the network.

These five strategies were examined in different aspects, including pathway-based disease state classification, PEA results, the fitness of the reconstructed BNs on non-tumor samples, and the plausibility of eliminated edges with the JAK-STAT signaling pathway. The BNs reconstructed by different methods all provide consistently high classification performance (AUC > 0.95); nonetheless, the pathways that played a major role in the classification were distinctly different. Since the reconstruction strategy is the only variable, these results suggest that there are tumor-influenced gene relationships that are inappropriately removed by some methods. This makes it urgent to dig deeper into the biological functions represented by edges preferentially removed by various methods. To this end, the canonical pathway JAK-STAT, which contains complex negative feedback loop structures, was analyzed. As expected, BNrich removed all edges representing negative feedback inhibition from SOCS to JAKs. The Ensemble method, which learns to infer and make decisions based on the graph hierarchy, also removes all edges from SOCS to JAK. This fits well with its core assumption that nodes receiving edges are ranked higher than nodes giving edges. In the case of biological pathways, genes such as JAKs and STATs that regulate a large number of downstream genes would be inferred by Ensemble as low-ranking, and thus edges pointing to them would be more likely to be dropped. Unlike BNrich and Ensemble, Clipper, another computational method that relies on a linear relationship between gene expression, preserves some negative feedback edges while removing more edges representing activated JAKs and/or cytokine receptor phosphorylation of STATs. This can be partially explained by the adaptive process, while maintaining feedback edges may also help to identify tumor-influenced inhibition. For the remaining two methods, PROPS and BPA, the random selection strategy makes them difficult to agree across different implementations, and edges representing key regulations are often omitted. As for BPA, while new edges are added randomly as a remedy, they are often excessive and can hinder true biological signaling representation.

In short, BNrich relies on expert knowledge to provide the most biologically logical BN. The only limitation is that manual selection is required every time a new pathway is considered. The Ensemble method prefers to remove edges pointing to key regulators with a large number of downstream genes, and this needs to be carefully inspected when used. The data-driven method, Clipper, produces similar yet different results to Ensemble, which can be partially explained from the perspective of biological principles to some extent. Therefore, Clipper can serve as a useful alternative when hand-crafted acyclic pathways are not available.

However, limitations still exist. Edge removal in Clipper and network simplification in BNrich are performed under the assumption of linear correlation, which is often not fulfilled in real scenarios. Some distribution-free methods, such as maximum local correlation and mutual information, should be incorporated in future research to detect the nonlinear relationships in the gene expression data [39]. The PROPS strategy is not recommended because randomness may hinder downstream analysis and network interpretation. For example, in the fitness of reconstructed BNs on non-tumor samples, PROPS consistently ranks in the bottom two. Edge addition in BPA is desirable as a remedy but doing so in a random fashion still loses important edges and can result in a large number of low-value relationships.

### 4.2. Suggestions for Future Research

One of the immediate future directions for developing BN reconstruction methods for pathway analysis is to combine the advantages of current methods, using BPA as a template since randomly removing edges that form cyclic structure can impair important biological causality. The strength of evidence for each edge inferred from expression data, performed in Clipper, can also be considered as a useful alternative. The idea of adding edges to preserve associativity is worth keeping, but the way of adding at random and connecting parent nodes to all loop members brings superfluous low-quality edges and does not guarantee the preservation of high-quality edges. The LASSO-based network simplification strategy used in BNrich can be used to refine newly added edges. In addition to linear relationships, nonlinear relationships should also be considered throughout. Finally, the biological intuitive rules-based approach such as that proposed in BNrich can serve as a guideline for checking and improving edge elimination and addition.

While we emphasize the comparison of methods based on an experimentally identified pathway, many existing BN methods for biological networks focus on reconstructing BN purely from expression data. They often do not take into account existing knowledge of experimentally determined pathways. Without pathway information as prior knowledge, they usually assume a uniform graph, which may be suboptimal. In addition, due to the computational complexity, they are often applicable to relatively small gene sets, whereas the volume of expression data generated by high-throughput platforms is often much larger. Therefore, the resulting high computational costs may limit the use of all available data and the reconstruction of complete network structures. Even so, their ability to infer underlying biological topology and causal relationships from the data and store them in BN in the form of edges is desirable for future study. One possible way to combine the context of known biological pathways with inference methods that learn structures from data is to perform Bayesian network structure refinement via carefully designed scores or constraints, starting from existing pathways. Given the high computational complexity resulting from a large number of genes, local learning may be preferred, which learns structures near one or more target genes that are of particular interest but not the rest.

## 5. Conclusions

In this study, we comprehensively evaluate and compare five representative BN reconstruction strategies among existing TPA methods. Through rigorous testing on five gene expression datasets of different cancer types, we show that while all methods perform well in disease state classification, their pathway rankings vary widely, indicating the sensitivity of the edge removal. The preserved edges within the complex cyclic structure of the JAK-STAT pathway were visualized and analyzed to reveal the differences among methods. Based on the domain knowledge of experts, BNrich provides network structures that best fit the biological facts. The gene expression data-driven approach, Clipper, may be a useful alternative, the results of which can also be explained to some extent by biological principles and theorems. Finally, the limitations and advantages of all methods are summarized to serve as a useful reference to facilitate future research.

## Figures and Tables

**Figure 1 biomolecules-12-00906-f001:**
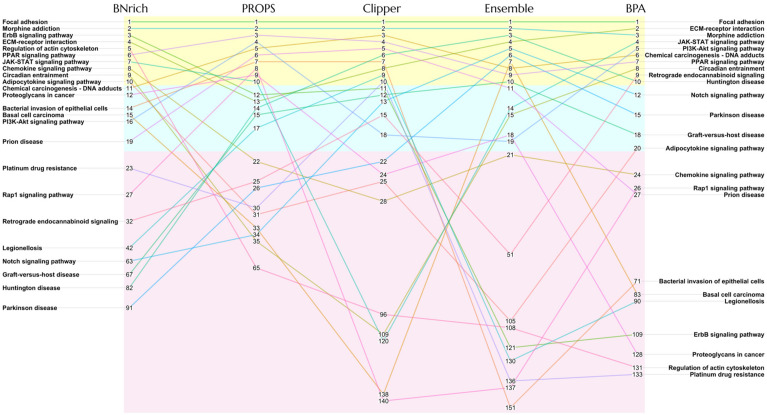
A bump chart summarizing the ranking of the top ten pathways enriched by any of the five BN reconstruction methods in HCC dataset. Numbers represent the PEA ranks of pathways produced by the corresponding BN reconstruction approach. Lines connecting the rank numbers provide an intuitive visualization of how pathways are changing in a ranking across five approaches.

**Figure 2 biomolecules-12-00906-f002:**
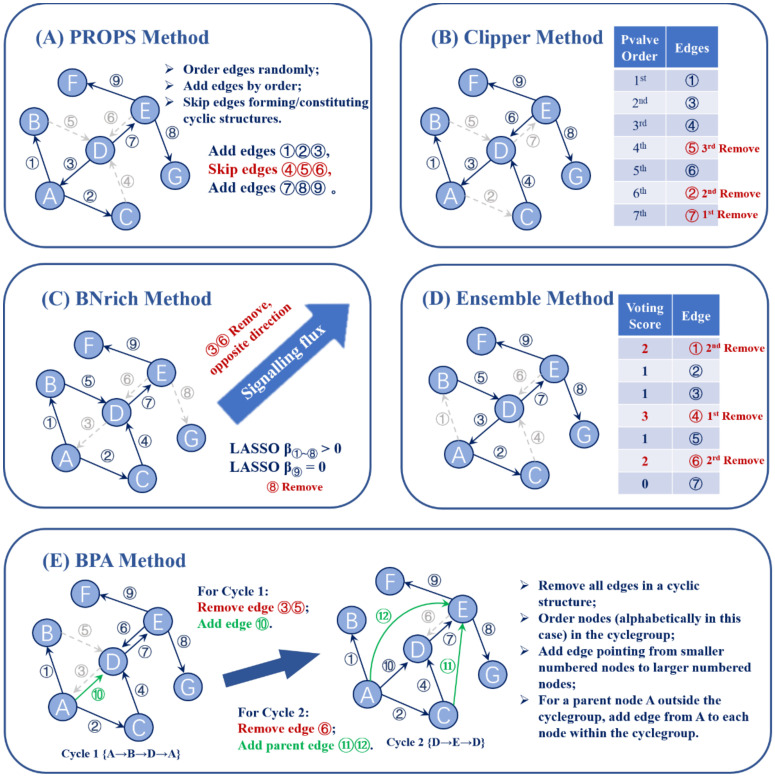
Schematic illustration of five BN reconstruction strategies. A hypothetical gene network, constituting 7 nodes (genes) and 9 edges is synthesized as an example to present the principle of each BN reconstruction method. Three cyclic structures {A -> B -> D ->A}, {A -> C -> D -> A}, and {D->E->D} are introduced in this network. (**A**) PROPS method orders each edge randomly and adds edges one by one. The edges that lead to cyclic structures are abandoned. Here, edge ④, edge ⑤, and edge ⑥ are removed. (**B**) Clipper method excludes cyclic edges with the most undesired *p*-value from the linear regression significance test. Edge ②, edge ⑤, and edge ⑦ are the lowest-ranked in three cyclic structures thus removed. (**C**) BNrich method firstly removes cyclic structures according to intuitive biological rules, herein edge ③ and edge ⑥ due to them being opposite in the direction of signaling transmission. Additionally, BNrich employs LASSO to further simplify the network. The coefficient of edge ⑨ is shrunken to zeros during LASSO regression, thus removed. (**D**) Ensemble method breaks cycles and preserves its graph hierarchy as much as possible. Edges in a cyclic structure with the highest voting score are excluded, Herein edge ①, edge ④, and edge ⑥ are removed to break corresponding cycles. (**E**) BPA method orders nodes arbitrarily (here in alphabetical order) and adds new edges from smaller numbered nodes to larger numbered nodes. Parent nodes of cyclic structure are also connected to each of the nodes in the cycle. Firstly edge ③ and edge ⑤ are removed and edge ⑩ is added in cycle 1. Then edge ⑥ is removed and node A and node C, two parent nodes of node D, are connected to node E constituting edge ⑪ and ⑫.

**Figure 3 biomolecules-12-00906-f003:**
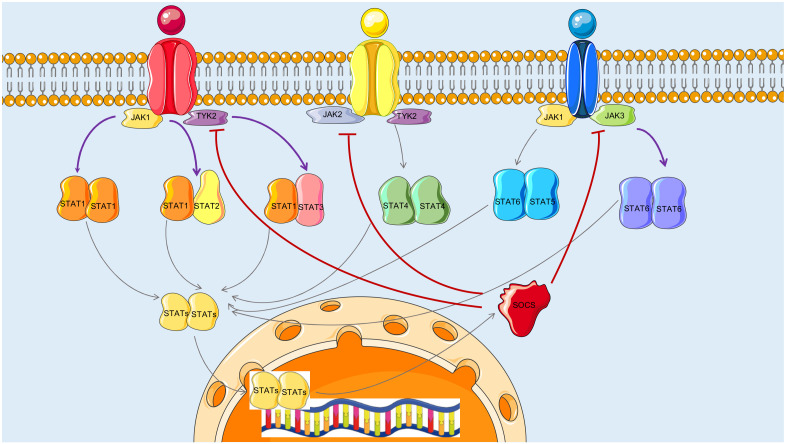
JAK-STAT-SCOS cell model. In canonical JAK-STAT signaling, the binding of extracellular cytokines to their transmembrane receptors results in activation of the pathway in a JAK-mediated manner. Both JAK and cytokine receptors are transphosphorylated by JAKs in close proximity. Activated JAKs phosphorylate STATs, leading to STAT dimerization and eventual translocation into cell nucleus, where they regulate the expression of genes, including SOCS proteins that act as negative feedback inhibitors of JAK-STAT. In BN reconstruction, both BNrich and Ensemble removed all the negative feedback edges representing SOCS inhibition of JAK-STAK (edges in red), while Clipper retained around 1/3 of them. In addition, Clipper also removed some of the JAK-induced STAT phosphorylation (edges in purple). Proteins of the STAT family are present in homodimers or heterodimers. SOCS family can be merged in a single node.

**Figure 4 biomolecules-12-00906-f004:**
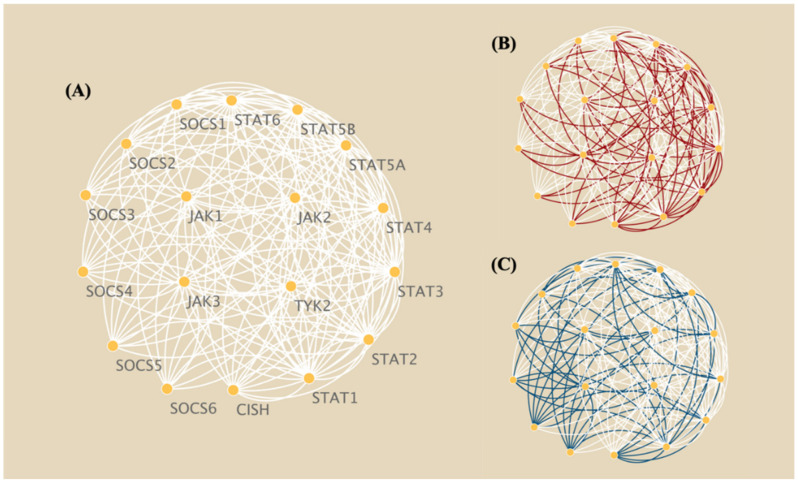
Simplified schematic diagrams of JAK-STAT-SOCS signaling and Clipper-reconstructed BNs from the original and randomly shuffled gene expression data. (**A**) Core paths of JAK-STAT signal transduction and SOCS negative feedback inhibition; (**B**) reconstructed BN (edges in red) from gene expression data of paired tumor and non-tumor HCC patients; (**C**) reconstructed BN (edges in dark blue) from shuffled gene expressions across samples.

**Figure 5 biomolecules-12-00906-f005:**
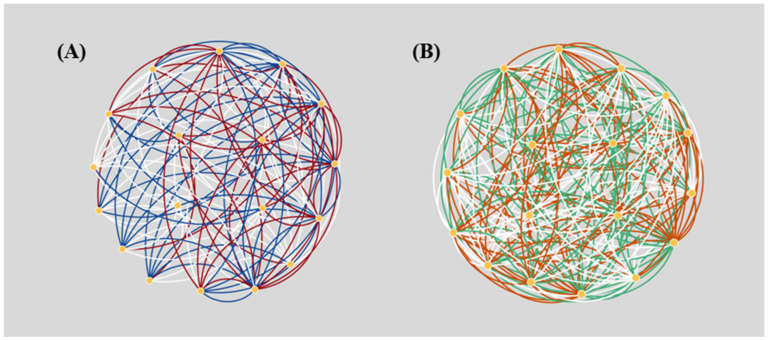
Schematic illustration of two superimposed PROPS (**A**) and BPA (**B**) randomly generated BNs of JAK-STAT-SOCS signaling. (**A**) Edges in red and blue represent those unique to the two PROPS-reconstructed BNs, respectively; (**B**) edges in orange and green represent those unique to the two BPA-reconstructed BNs, respectively. The low coincidence of the edges contained in the two networks can be observed, indicating the randomness of the BN structures by the two approaches.

**Table 1 biomolecules-12-00906-t001:** Summary of representative PEA approaches and their applications.

Method	Type	Example Application	Data Type	Year	Ref.
GoMiner	ORA	Compare gene expression profiles in a prostate cancer cell line and a subline selected from it for resistance to the topoisomerase 1-inhibitor 9-nitro-camptothecin.	Microarray	2003	[3]
WebGestalt	ORA	Disease association analysis and drug association analysis in colorectal cancer; Elucidation of the pharmacological mechanisms of Rhynchophylline for treating epilepsy.	Microarray; RNA-seq	2013	[4,8]
GSEA	FCS	Searched for significantly associated gene sets from: male vs. female lymphoblastoid cells; p53 status in cancer cell lines; acute lymphoid leukemia (ALL) vs. acute myeloid leukemia (AML); comparing two studies of lung cancer.	Microarray; RNA-seq	2005	[5]
BNrich	TPA	Discriminate among specific experimental states in: tumor cell line skin, tumor cell line large intestine, breast cancer, Alzheimer, colorectal cancer.	Microarray	2020	[9]
Clipper	TPA	Peak calling from ChIP-seq data, peptide identification from MS data, DEG identification from bulk or single-cell RNA-seq data, and DIR identification from Hi-C data.	ChIP-seq; microarray; RNA-seq; mass spectrometry	2012	[10]
PROPS	TPA	Classification of inflammatory bowel disease.	Microarray	2017	[11]
TopologyGSE	TPA	Identify key regulatory elements in gene expression data on (i) acute lymphocytic leukemia (ALL) with and without BCR/ABL gene rearrangement and (ii) lung adenocarcinoma with and without EGFR mutation.	Microarray	2010	[12]
BPA	TPA	Investigated the active pathways in NCBI’s GEO database regarding bladder, brain, breast, colon, liver, lung, ovarian, and thyroid cancers.	Microarray	2011	[13]
BAPA-IGGFD	TPA	Understand the gene expression profile of osteoblast lineage at defined stages of differentiation.	Microarray	2012	[14]
Ensemble	TPA	Breaking cycles while preserving the logical structure of the directed graph.	Directed graph	2017	[15]

**Table 2 biomolecules-12-00906-t002:** Tumor and normal tissue classification performance based on pathways reconstructed by BNrich, PROPS, Clipper, BPA, and Ensemble method. The thresholds for accuracy, recall, precision, and F1-score were set to be the one that obtains the best MCC score.

Dataset	Method	Accuracy	Recall	Precision	F1-Score	AUC
HCC	BNrich	0.95 ± 0.04	0.95 ± 0.03	0.95 ± 0.06	0.95 ± 0.03	0.96 ± 0.04
	PROPS	0.93 ± 0.05	0.93 ± 0.06	0.94 ± 0.07	0.93 ± 0.04	0.95 ± 0.04
	Clipper	0.93 ± 0.04	0.92 ± 0.06	0.94 ± 0.04	0.93 ± 0.04	0.95 ± 0.04
	BPA	0.92 ± 0.06	0.93 ± 0.09	0.91 ± 0.07	0.92 ± 0.06	0.95 ± 0.05
	Ensemble	0.93 ± 0.05	0.90 ± 0.08	0.96 ± 0.06	0.93 ± 0.05	0.96 ± 0.05
HNSC	BNrich	0.93 ± 0.05	0.91 ± 0.12	0.96 ± 0.07	0.93 ± 0.06	0.97 ± 0.03
	PROPS	0.94 ±0.05	0.96 ± 0.09	0.92 ± 0.08	0.94 ± 0.05	0.97 ± 0.03
	Clipper	0.93 ± 0.06	0.94 ± 0.12	0.93 ± 0.08	0.93 ± 0.07	0.95 ± 0.04
	BPA	0.95 ± 0.04	0.98 ± 0.04	0.93 ± 0.07	0.95 ± 0.04	0.96 ± 0.03
	Ensemble	0.95 ± 0.04	0.97 ± 0.04	0.95 ± 0.07	0.96 ± 0.03	0.96 ± 0.04
KIRC	BNrich	0.99 ± 0.00	0.99 ± 0.00	0.99 ± 0.00	0.99 ± 0.00	0.99 ± 0.00
	PROPS	0.98 ± 0.01	0.98 ± 0.03	0.99 ± 0.00	0.98 ± 0.02	0.99 ± 0.00
	Clipper	0.98 ± 0.01	0.98 ± 0.03	0.99 ± 0.00	0.98 ± 0.02	0.99 ± 0.00
	BPA	0.99 ± 0.00	0.99 ± 0.00	0.99 ± 0.00	0.99 ± 0.00	1.00 ± 0.00
	Ensemble	0.99 ± 0.00	0.99 ± 0.00	0.99 ± 0.00	0.99 ± 0.00	1.00 ± 0.00
LUAD	BNrich	0.99 ± 0.00	0.98 ± 0.04	1.00 ± 0.00	0.99 ± 0.00	1.00 ± 0.00
	PROPS	0.99 ± 0.00	0.98 ± 0.04	1.00 ± 0.00	0.99 ± 0.00	1.00 ± 0.00
	Clipper	0.99 ± 0.00	0.98 ± 0.04	1.00 ± 0.00	0.99 ± 0.00	1.00 ± 0.00
	BPA	1.00 ± 0.00	0.99 ± 0.00	1.00 ± 0.00	1.00 ± 0.00	1.00 ± 0.00
	Ensemble	0.99 ± 0.00	0.98 ± 0.03	1.00 ± 0.00	0.99 ± 0.00	1.00 ± 0.00
THCA	BNrich	0.96 ± 0.04	0.98 ± 0.04	0.95 ± 0.07	0.97 ± 0.03	0.98 ± 0.02
	PROPS	0.96 ± 0.04	0.98 ± 0.04	0.95 ± 0.07	0.96 ± 0.04	0.98 ± 0.03
	Clipper	0.96 ± 0.04	0.99 ± 0.00	0.94 ± 0.07	0.96 ± 0.04	0.98 ± 0.03
	BPA	0.96 ± 0.04	0.99 ± 0.00	0.94 ± 0.06	0.97 ± 0.03	0.97 ± 0.03
	Ensemble	0.96 ± 0.04	0.99 ± 0.00	0.94 ± 0.07	0.96 ± 0.04	0.97 ± 0.03

**Table 3 biomolecules-12-00906-t003:** Averaged ranks of BIC scores of all pathways by BPA, Clipper, PROPS, BNrich, and Ensemble method across five datasets. Values in bold indicate the best ranked method in specific datasets.

Method	Averaged Ranks of BIC Scores				
	HCC	HNSC	THCA	KIRC	LUAD
BPA	**2.03**	3.32	**2.34**	**2.36**	2.74
Clipper	3.03	3.20	2.59	2.59	2.65
PROPS	3.45	3.84	3.77	3.61	3.56
BNrich	2.78	2.95	2.76	2.74	**2.51**
Ensemble	3.70	**1.69**	3.54	3.69	3.54

## Data Availability

RNA sequencing data generated during this study are included in this published article and its Appendix A. HCC dataset (GSE144269) is freely available from the public database NCBI Gene Expression Omnibus (GEO), and HNSC, KIRC, LUAD, and THCA datasets in FPKM are freely available from The Cancer Genome Atlas (TCGA) research network.

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
