# Peer review of "Comparing Bayesian-Based Reconstruction Strategies in Topology-Based Pathway Enrichment Analysis"

_biomolecules, 2022, doi:10.3390/biom12070906_

Round 1

Reviewer 1 Report

The quality of the paper is good. Overall the structure and the writing of the paper are also good. Nevertheless, there are some points that the authors must address:

  1. A key aspect of BN reconstruction is the identification of cycles. For example, line 6 of Algorithm 1 (function cyclic (G)) or line 13 of Algorithm 2. What method/algorithm is used to identify if a Graph has a cycle or not? Authors should describe this since it is an essential part of BN reconstruction.
  2. Algorithm 2 and Algorithm 3 use linear regression to identify relations between nodes (Algorithm 3 with lasso). Nevertheless, this approach can only identify linear relations. What happens with the non-linear relations? Authors should comment about this limitation.
  3. In subsection 2.5 it is stated that accuracy and AUC values would be reported. Nevertheless, AUC results are never reported. Also, in addition to accuracy and AUC, I would suggest that the authors include Precision, Recall, and F1-score.

Reviewer 2 Report

The research paper deals with the pathway enrichment analysis (PEA), which provides an intuitive solution for extracting biological insights from massive amounts of data. Moreover, topology-based pathway analysis (TPA) represents the latest generation of PEA methods which exploit pathway topology in addition to lists of differentially expressed genes and their expression profiles. A subset of these TPA methods, such as BPA, BNrich, and PROPS, reconstruct pathway structures by training Bayesian Networks (BNs) from canonical biological pathways, providing superior representations that explain causal relationships between genes. However, these methods had never been compared for their difference in the PEA and their different topology reconstruction strategies. In this paper, authors aim to compare the BN reconstruction strategies of BNrich, PROPS, and BPA and their PEA performance in hepatocellular carcinoma (HCC) and non-HCC classification based on gene expression data.

This seems to be a very challenging topic and authors have solidly proven their findings. Nevertheless, some points need to be clarified before this submission can be accepted.

The introduction seems inadequate in terms of the discussion of the problem at hand as well as the contribution subsection must be included in order to emphasize the problem, the goal as well as general remarks regarding the results.

The authors cannot efficiently state the differences among their work and the references.

There is no methodology or proposed method section. Just the very useful materials.

Results section consists of two tables with numerical values. What authors try to prove with these experiments? How are they differentiated from other works?

The discussion section is based on these implementations. Maybe, authors try to validate more in detail these structures in terms of their weaknesses in order for their paper to be more comprehensive.

Finally, there is no conclusions section. Please elaborate. Just the discussion which summarizes the results of the experiments.

The same stands for the title of the last section. The last paragraph deals with future work but there is no title containing the phrase “future work”.

Reviewer 3 Report

I have to state that I have a positive opinion about this study in consequence of its importance. However, I would like to draw the attention of the authors to the important points that need to be corrected in the article. Before acceptance, the following points must be incorporated.

1.      Report the results in abstract, there is a need to report the quantitative results in the abstract.

2.      The objective, motivation, and research gap is unclear.

3.      The contribution from the authors does not look novel: referred to the An investigation of credit card default prediction in the imbalanced datasets.

4.      The literature survey section should be separated with heading: referred to the A novel framework for prognostic factors identification of malignant mesothelioma through association rule mining.

5.      Comparison of previous researches must be done as in table 1 of referred paper: Corporate Bankruptcy Prediction: An Approach Towards Better Corporate World. In table 1, there is need to report the dataset characteristics.

6.      Add also latest applications of computational methods in the similar domain. Consult the following papers:

Sparse PLS-Based Method for Overlapping Metabolite Set Enrichment Analysis

Early Prediction of Malignant Mesothelioma: An Approach towards Non-invasive Method

EcoDiet: A hierarchical Bayesian model to combine stomach, biotracer, and literature data into diet matrix estimation

Disease diagnosis system using IoT empowered with fuzzy inference system

7.      The author’s justify and explain the evaluation criteria.

8.      Conclusion must be included.

9.      Discussion section should be summarized.

10.  The author’s justify and explain why proposed model achieved such high results.

11.  Summarize the experimental results with some numeric vales in the discussion sufficiently.

12.  It is better to summarize the major findings of the paper within one-two sentences without experimental results.

Overall speaking, the innovation points and main contributions of this paper need to be carefully reconsidered, and the innovation points should be presented more clear and prominent in terms of word expression and methodology & experiment design.

Round 2

Reviewer 1 Report

My previous comments have been addressed satisfactorily in the revised version.

Reviewer 2 Report

Authors have addressed my comments so I vote for acceptance

Reviewer 3 Report

Accepted